# Toddlers’ Language Development: The Gradual Effect of Gestational Age, Attention Capacities, and Maternal Sensitivity

**DOI:** 10.3390/ijerph17217926

**Published:** 2020-10-29

**Authors:** Vera E. Snijders, Lilly Bogicevic, Marjolein Verhoeven, Anneloes L. van Baar

**Affiliations:** Department of Clinical Child and Family Studies, Utrecht University, 3584 CS Utrecht, The Netherlands; l.bogicevic@uu.nl (L.B.); j.c.t.verhoeven@uu.nl (M.V.); a.l.vanbaar@uu.nl (A.L.v.B.)

**Keywords:** receptive language, expressive language, moderate prematurity, attention, maternal sensitivity

## Abstract

Language development in toddlerhood forms the foundation for speech and language comprehension throughout childhood. Children born moderately preterm are at increased risk for problems in receptive and expressive language functioning, and they may need specific support or interventions. To understand the underlying mechanisms of language development, an integrated model of gestational age, attention capacities, and maternal sensitivity was examined in relation to receptive and expressive language functioning in toddlerhood. Our sample included 221 children (gestational age between 32–41 + 6 weeks; 54.7% born moderately preterm; 51.6% boys; 69.1% highly educated mothers). At 18 months (corrected age), attention capacities were measured using an eye-tracking procedure and maternal sensitivity was observed during mother-child interaction. Language was assessed at 24 months (corrected age). Results showed children with a higher gestational age scored higher on receptive language. This association was direct, as well as indirect through the child’s alerting attention. Expressive language was related to maternal sensitivity. Gestational age and alerting attention capacities specifically were related to language comprehension, whereas maternal sensitivity was related to speech. As language comprehension and speech in toddlerhood show different associations with biological, child, contextual, and regulation factors, they should be viewed as separate constructs in research and practice.

## 1. Introduction

Language forms the means to understand and communicate with others, which is key to social interaction. Moreover, early language development forms the basis for language functioning throughout childhood. Previous research indicates considerable stability in children’s language development from 20 months to 8 years [1], suggesting that the quality of language development during the first two years of life predicts development of language functioning throughout childhood. In order to foster healthy language development, it is key to understand which factors promote this process of language development. Early detection of difficulties and potential risk factors of problems in language development can enable a focused design and implementation of intervention programs aimed at preventing further problems in language development.

As a general framework to study child development, we use Sameroff’s Unified Theory of Development [2]. This theory states that human development can be understood from at least four models: The personal change (i.e., psychological as well as underlying biological experiences and changes within the individual), regulation (i.e., self- and other-regulation processes), contextual (i.e., the child’s environment on different levels, as proposed by Bronfenbrenner [3]), and representational model (i.e., meanings an individual creates regarding here and now experiences in the world). Sameroff [2] unifies these four models into an integrated framework. In the current study, language functioning is examined in relation to a combination of potentially important risk and mediating factors reflecting different parts of this framework: prematurity as a biological risk factor in the personal change model, children’s attention capacities showing personal characteristics, and mother-child interaction quality showing contextual and regulation aspects.

Prematurity is a biological risk factor for suboptimal language functioning. Children born preterm (i.e., <37 weeks of gestation) are at risk for problems in both simple and complex language functioning, even in the absence of major disabilities [4]. Children born preterm have had less time for brain maturation within the womb and are at increased risk of neonatal complications compared with children born full-term [5,6]. These early experiences may affect how the preterm born child’s brain develops, and consequently, influence development of competencies such as language. The majority of preterm births are accounted for by children born moderately preterm (i.e., 32–36 + 6 weeks of gestation; 87.4%; [7]). Indeed, it has already been shown that children born moderately preterm have more problems in language functioning than children born full-term at toddler age, both in receptive and expressive language [1,8,9,10,11,12,13]. Even children born early term (i.e., 37–38 + 6 weeks of gestation; [12]) score lower on language functioning.

Important is also that gestational age (from 25–42 weeks) has been found to show a gradual effect on child language functioning [1]. Moreover, cognitive assessments based on chronological age deeply underestimate developmental status of the child in the first years of life [14]. By using corrected age scores (i.e., age corrected for the weeks a child is born premature), the differences between full-term and preterm born children in immature development are eliminated. Therefore, we will study how language functioning, corrected for prematurity, is gradually affected by gestational age in children born moderately preterm or full-term.

If we aim to understand which factors may explain why children born at a lower gestational age are at higher risk to experience problems in language functioning, we also need to consider children’s elementary cognitive skills, such as attention capacities. Attention is a multidimensional construct that can be separated into three domains: orienting (i.e., ability to engage, disengage, and shift attention), alerting (i.e., ability to achieve and maintain a state of alertness), and executive attention (i.e., goal-directed and planned attention, or attentional control) [15,16]. Elementary cognitive skills—such as attention capacities—are core skills in learning to understand and produce language [17,18]. Specifically, toddlers who have better attention capacities also often have better language skills, both for children born preterm [19,20] and full-term [21,22]. Moreover, an increased risk for deficits in orienting, alerting, and executive attention has been demonstrated in children born preterm at 18 months [23,24], underlining the importance of attention capacities in language development of children born at a lower gestational age.

To date, few studies have assessed whether attention capacities mediate the relation between gestational age and language functioning in preterm born samples. An exception to this is the study by De Schuymer and colleagues [25], demonstrating that deficits in obtaining attention and joint attention (i.e., both part of the orienting attention system) at 14 months were related to receptive and expressive language at 30 months in children born very preterm. Obtaining attention mediated the relation between gestational age and receptive and expressive language, while joint attention only mediated the relation between gestational age and receptive language. This study indicates that engagement and attention in interaction with a caregiver is an essential part in developing healthy language functioning. Furthermore, Mahurin-Smith and colleagues [26] showed that gestational age was associated with attention capacities at 7 years, and attention capacities were related to language functioning at 10 years of age in children born very preterm and full-term. In both studies, gestational age was also directly related to language functioning in addition to the indirect effect [25,26], indicating attention is not the only mechanism that mediates the relation between gestational age and language functioning.

Socioenvironmental factors, as part of the contextual model [2], also support language functioning. How the mother behaves in interaction with her child and the content and quality of communicative experiences she provides are such socioenvironmental factors that affect the child’s language development [27]. Specifically, previous research has shown that toddlers whose mothers were more sensitive in interaction (i.e., mother’s warmth, responsiveness) had better expressive [28,29,30] and receptive language skills [29,30], indicating that maternal sensitivity is an important factor promoting healthy language development of children born preterm. In the mother-child relationship, maternal behavior in interaction, specifically maternal sensitivity, may be different for children born preterm and children born full-term [19]. Results from previous studies on the association between prematurity and maternal sensitivity have been mixed. Some studies found no difference in sensitivity between mothers of preterm and mothers of full-term children [19,31], while other studies found mothers of preterm children to be more sensitive [32] or less sensitive [31,33] than mothers of full-term children. While maternal sensitivity has not been assessed as a mediator in children born moderately preterm, Loi and colleagues [19] found that children born very and extremely preterm had higher risk of language delay when their mothers were less sensitive during interaction. 

Given that environmental factors are important for children’s cognitive skills according to the regulation model [2], we also need to consider the relation between maternal sensitivity and attention capacities. Previous research indicates that maternal sensitivity is related to attention capacities, as lower maternal sensitivity was related to poorer attentional control (i.e., part of the executive attention system) at 6 and 8 years [34], and sustained attention (i.e., part of the alerting attention system) at 2 years of age [35] in children born full-term. These studies indicate that maternal sensitivity supports the development of attention in early childhood. We therefore also consider the relation between maternal sensitivity and attention capacities.

In the current study, language functioning is examined in relation to different risk and mediating factors at toddler age (see Figure 1). Language functioning was studied in relation to biological risk factors (i.e., gestational age), children’s attention capacities, and maternal sensitivity. Based on previous research, we expected that (a) higher gestation at birth would be associated with better receptive and expressive language functioning at 24 months; (b) attention capacities and maternal sensitivity partially mediate this relation; and (c) maternal sensitivity would also partially mediate the relation between gestational age and attention capacities.

## 2. Materials and Methods 

### 2.1. Participants

Participants in this study are part of a prospective longitudinal study on the development of children born moderately preterm in comparison to children born full-term: The Study of Attention in Premature children (STAP) Project. Children included in this project were born in nine hospitals in and around Utrecht in the Netherlands between March 2010 and April 2011. All children have a gestational age between 32 and 41 + 6 weeks. Exclusion criteria included admission to a tertiary Neonatal Intensive Care Unit, birth weight below the 10th percentile according to Dutch references [36], severe congenital malformations, multiple births, antenatal alcohol or drug abuse and chronic antenatal use of psychiatric drugs by the mother. 

The project initially contained a sample of 226 children. At 18 months of corrected age (i.e., age corrected for prematurity), 214 children (94.7%) and their mothers were observed during mother-child interaction in a lab setting. An eye-tracking procedure, focusing on attention capacities, was performed for 207 children (91.6%). At 24 months of corrected age, 214 children (94.7%) provided data on language functioning. Children included in the current study provided data on at least one of the assessments, resulting in a sample of 221 children (97.8% of the original sample). Within the final sample, 121 children were born moderately preterm (gestational age in weeks: *M* = 34.68, *SD* = 1.34; birth weight in grams: *M* = 2589.71, *SD* = 515.86; days in hospital: *M* = 11.81, *SD* = 9.84), and 100 children were born full-term (gestational age in weeks: *M* = 39.48, *SD* = 0.98; birth weight in grams: *M* = 3576.34, *SD* = 458.09; days in hospital: *M* = 0.41, *SD* = 1.01). All parents could read and speak Dutch. No differences were found in demographic and neonatal characteristics between children included in analysis (*n* = 221) and those who dropped out (*n* = 5). Participants’ demographic and neonatal characteristics are presented in Table 1. 

### 2.2. Procedure

The STAP Project was approved by the medical ethical committee of the Utrecht Medical Center. Data used in this study were collected between March 2011 and March 2013. Parents of the children were invited to participate when the children were 10 months old. At 18 months corrected age, children visited the lab for an eye-tracking procedure assessing attention capacities for around 18 min [37], and observation of a mother-child interaction for 15 min. For the large majority children, the eye-tracking procedure preceded the mother-child interaction. At 24 months corrected age, the children visited the lab or nearby hospital for language assessment, which took about 83 min on average (*M* = 82.91, *SD* = 12.44). Informed consent was obtained from parents. After each visit, the children received a small gift and parents’ travel expenses were refunded.

### 2.3. Measures

#### 2.3.1. Attention Capacities

Attention capacities were assessed at 18 months corrected age with the Utrecht Tasks of Attention in Toddlers using Eye tracking (UTATE; [37]). The UTATE include four tasks: (1) disengagement task, (2) face task, (3) alerting task, and (4) delayed response task. In the disengagement task, a visual stimulus was presented in the center of the screen, followed by a second stimulus after 2 s, either left or right of the first stimulus. This task included 20 trials. In the face task, two identical pictures of a child’s face were shown. After 8.5 s, one of the pictures changed into a new picture, and these two pictures were then shown for another 8 s. This task consisted of 8 trials. In the alerting task, a visual stimulus was presented on the screen in 32 trails. In half the trails, this was preceded by a signaling sound. Finally, in the delayed response task, a voice-over pointed the child toward a dog on the screen and told the child the dog wants to play “hide-and-seek”. The child was told to pay attention, because the dog is going to hide himself. The dog moves to one of the doghouses for 1000 milliseconds, before he disappears. A worm pops up in the center of the screen to distract the child from the dog houses, and after a delay, the voice-over asked the child to search for the dog. This task consisted of 18 trials, in which the delay increased with 2 s after three trials, from 0 to 10 s [37]. From these four tasks, multiple variables were derived, which were then used to form latent factors of orienting, alerting, and executive attention [23]. The factor structure of these variables is described in detail elsewhere [38]. Split-half reliability for orienting, alerting, and executive attention was good [23].

#### 2.3.2. Maternal Sensitivity

At 18 months (corrected age), a 15-min mother-child interaction was observed. The mother-child dyads first participated in 5 min of free play with a variety of toys, followed by two times 5 min of structured play (i.e., reading a book and making a puzzle). The mother-child interaction was videotaped and coded by nine trained coders blinded for gestational age. Coders had a reliability score of 0.70 after training, and interrater reliability was good with an intraclass correlation of 0.76 based on 21% double-coded videos used in this study.

Interactions were coded using the Coding Interactive Behavior (CIB) system [39]. This system includes 42 codes rated on a 5-point scale ranging from low to high, which are then aggregated into several composite scores. The CIB has been validated in studies of healthy and at-risk dyads, has good psychometric properties, and has shown sensitivity to variation in infant age and biological and social-emotional risk factors [40,41,42]. In the current study, only codes for maternal sensitivity were used. A composite score was computed as indicated in the manual [39], which included seven items, coded separately in the free play and structured play conditions (i.e., acknowledgement of child communications, vocal clarity, positive affect, appropriate range of affect, resourcefulness, adaptation to child signals, and supportive presence). A higher composite score indicated higher maternal sensitivity. We performed reliability analysis for both composite scores in the full-term (i.e., healthy) group to see whether it was more reliable to create separate composite scores for the free play and structured play condition, or to include free and structured play in one composite score. The Cronbach’s alpha was highest in the composite scores that included items of both free and structured play (α = 0.81).

#### 2.3.3. Language Functioning

At 24 months (corrected age), The Bayley-III-NL [43] was administered by trained examiners, blinded to the children’s background, to assess children’s developmental level. The Bayley-III-NL consists of five subtests: Cognition, fine motor, gross motor, receptive communication, and expressive communication. In the current study, receptive and expressive communication scales were used. The receptive communication scale is intended to assess auditive skills, and the understanding of words and questions that result in an adequate response. As such, the receptive communication subtest addresses preverbal behavior, vocabulary development (i.e., recognizing objects, morphological development), and referrals in social interaction. Examples of items include identifying pictures and objects. The expressive communication scale is intended to assess the vocalizations of younger children, and the expressive language of older children. As such, the expressive communication subtest addresses preverbal communication, vocabulary development, and morpho-syntactic development. Examples of items are combining words and gestures, and naming objects. Dutch norms with good reliability and validity were used [43].

### 2.4. Data Analyses

Data were prepared in IBM SPSS Statistics 25 and further analyzed in Mplus 8.3 using Full Information Maximum Likelihood (FIML; [44]). Pearson’s correlations were used to examine univariate relations between the variables in the model. Direct and indirect effects were tested using structural equation modelling. Three separate models were run for each attention subdomain, as a previous confirmatory factor analysis showed that a three-factor model according to the theoretical model of attention [15] fit the data best [37,38]. The models were ran using normed scores for language functioning (corrected age). Moreover, models were adjusted for maternal educational level, birth order (i.e., ranging from first-born to fifth-born) and gender, as previous literature showed that these factors may be associated with early language development [45,46,47]. Because gestational age was not equally distributed in the data, all models were bootstrapped, and the 95% bias-corrected bootstrapped confidence intervals (CIs) were used as indicator of significance of the coefficients and indirect effects in the model [48]. The use of CIs is also recommended in assessing indirect effects [49,50]. Model fit was assessed using the chi-square model test statistic (χ^2^), the root mean square error of approximation (RMSEA), the comparative fit index (CFI) and Tucker–Lewis index (TLI). Model fit was good when the *p*-value of χ^2^ > 0.05, RMSEA < 0.06, CFI > 0.95, and TLI > 0.90 [51]. In case of removing non-significant control variables, different nested models were compared to each other using the Bayesian Information Criterion (BIC) value, where the lowest BIC-value indicated the best-fitting model [52].

## 3. Results

### 3.1. Descriptive Statistics and Correlations between Model Variables

The mean score on the language tasks of the Bayley-III-NL (corrected age) in the present sample was above average (*M* = 107.77, *SD* = 12.39, Range = 71–138). Thirteen children scored < 85 (6.1% of the total sample), indicative of mild developmental delay. The means, standard deviations, and intercorrelations of the variables in the model are presented in Table 2. 

### 3.2. Model Results

#### 3.2.1. Orienting Attention

The model for orienting attention, adjusted for maternal educational level, gender, and birth order, showed a reasonable fit (Table 3, Model 1a). Because maternal educational level was not significantly related to receptive language (95% CI (−0.336, 0.406)) and expressive language (95% CI (−0.203, 0.587)), we investigated whether model fit improved for a more parsimonious model adjusted only for gender and birth order. This model showed good fit (Table 3, Model 1b), fitting the data better than Model 1a (BIC_adjusted for gender and birth order_ = 2455 < BIC_+ adjusted for maternal educational level_ = 2464). While birth order was not significantly related to receptive language, there was a significant association with expressive language (Table 4). Therefore, the model adjusted for gender and birth order (Model 1b) is interpreted. Parameter estimates are reported in Table 4. Lower gestational age was significantly related to lower orienting attention capacities and lower receptive language functioning. Orienting attention capacities were not directly related to receptive or expressive language functioning. Lower maternal sensitivity was significantly related to lower expressive language functioning. None of the indirect effects were significant. In this final model (Figure 2a) gestational age, maternal sensitivity, and orienting attention capacities explained 10.5% of the variance in receptive language and 10.4% of the variance in expressive language.

#### 3.2.2. Alerting Attention

For alerting attention, the model adjusted for maternal educational level, gender, and birth order showed good fit (Table 3, Model 2a). Moreover, in this model, maternal educational level was not significantly related to receptive language (95% CI (−0.110, 0.151)) or expressive language (95% CI (−0.069, 0.209)). A more parsimonious model, adjusted only for gender and birth order, had good fit (Table 3, Model 2b), fitting the data better than Model 2a (BIC_adjusted for gender and birth order_ = 2591 < BIC_+ adjusted for maternal educational level_ = 2601). As birth order showed a significant relation with expressive language (Table 4), it was left in the model and the model adjusted for gender and birth order (Model 2b) is interpreted. Parameter estimates are reported in Table 4. Lower gestational age was significantly related to lower alerting attention capacities and lower receptive language functioning. Lower alerting attention capacities were significantly related to lower receptive language functioning. Maternal sensitivity was not significantly related to any parameter in the model, including expressive language, although the effect size was similar to the other models (β = 0.137). The indirect effect from gestational age to receptive language functioning through alerting attention capacities was significant (β = 0.031), with a mediation proportion of 17.6%. This indicates that alerting attention explains a substantial part of the direct relation between gestational age and receptive language functioning. In this final model (Figure 2b), gestational age, maternal sensitivity, and alerting attention capacities explained 11.6% of the variance in receptive language and 10.6% of the variance in expressive language.

#### 3.2.3. Executive Attention

For executive attention, the model adjusted for maternal educational level, gender, and birth order showed good fit (Table 3, Model 3a). Again, as maternal educational level was not significantly related to receptive language (95% CI (−0.278, 0.537)) or expressive language (95% CI (−0.170, 0.690), we investigated whether fit improved for a more parsimonious model, adjusted only for gender and birth order. This model also showed a good fit (Table 3, Model 3b), fitting the data better than Model 3a (BIC_adjusted for gender and birth order_ = 2660 < BIC_+ adjusted for maternal educational level_ = 2669). Again, birth order was left in the model as a control variable, as it showed a significant relation with expressive language (Table 4). The model adjusted for gender and birth order (Model 3b) is interpreted. Parameter estimates are reported in Table 4. Lower gestational age was significantly related to lower receptive language functioning. Lower maternal sensitivity was related to lower expressive language functioning. Executive attention was unrelated to any parameters in the model. In this final model (Figure 2c), gestational age, maternal sensitivity, and executive attention capacities explained 9.6% of the variance in receptive language and 10.2% of the variance in expressive language.

## 4. Discussion

In the current study, potential risk and mediating factors regarding language functioning at 24 months were examined in an integrated model that included a biological risk factor (lower gestational age), elementary cognitive functioning (attention capacities), and quality of maternal behavior during mother-child interaction (maternal sensitivity). We hypothesized that (a) gestational age would be related to receptive and expressive language functioning, (b) attention capacities and maternal sensitivity would mediate these relations, and (c) maternal sensitivity would also mediate the relation between gestational age and attention capacities. We found that gestational age was positively, directly related to receptive language and orienting and alerting attention; gestational age was indirectly related to receptive language functioning through alerting attention capacities; and maternal sensitivity was positively directly related to expressive language functioning.

Correcting for gender and birth order, we found partial support for hypothesis (a): While children born at a lower gestational age showed poorer receptive language functioning, their expressive language functioning was not significantly poorer at 24 months of (corrected) age. Comparing our results to other studies, we also found support for the fact that children born at a lower gestational age have suboptimal language functioning [1,8,9,11,12,13,53]. However, most of these studies examined language functioning as one construct, instead of differentiating between expressive and receptive language functioning [1,9,12,53], or included children at even higher risk, i.e., born at very low gestational ages (<32 weeks) [11,53]. Our findings also provide further evidence for a gradual effect of gestational age on receptive language [1,12], given that children born at lower gestational ages showed poorer receptive language. 

As expected, children born at a lower gestational age showed poorer receptive language as explained by poorer alerting attention. However, orienting and executive attention did not explain the relation between gestational age and receptive language. Our finding thus indicates alerting attention to be most important for early language development. Two possible explanations underlying the role of alerting attention in language functioning in children born at a lower gestational age are considered. Toddlers born at a lower gestational age are more likely to experience problems maintaining their focus and may in turn be less attentive [54] to the linguistic input from their environment. This could affect their ability to learn to understand language. However, toddlers born at a lower gestational age with poorer alerting attention may also have more trouble maintaining focus on and completing long assessments (e.g., Bayley-III-NL). Consequently, they obtain lower scores, which would then reflect attention problems rather than poorer language functioning. However, beyond the mediating effect of alerting attention, we also found that lower gestational age was directly related to poorer receptive language functioning. Thus, attention problems in a testing situation could only partially account for lower language scores on standardized language assessment. Therefore, it is more likely that poorer alerting attention skills affect the ability of a child to internalize language, i.e., receptive language as described in the first explanation. 

Orienting and executive attention, however, did not mediate the relation between gestational age and language functioning. Children born at a lower gestational age showed poorer orienting attention capacities, in line with previous research on this attentional subdomain [25,55]. These lower orienting attention capacities, however, did not affect language functioning at a later age. While previous research in very preterm samples did find orienting attention deficits to predict poorer language functioning [25], our sample was constituted of relatively healthy children born moderately preterm. This may have made our sample more similar to full-term samples in some aspects, in which a relation between orienting attention and language is not necessarily found [56]. Moreover, alerting attention provides children with the unique opportunity to sustain attention during word learning events, such as naming objects [57], which facilitates word learning. Orienting attention may merely provide the child with the opportunity to selectively attend to such events but not specifically to the word learning that occurs within the attentive period [22,57]. Regarding executive attention, we did not find relations with gestational age nor language functioning. The absence of a relation between gestational age and executive attention is in line with previous research by Voigt et al. [24], who found a relation for very preterm but not for moderately preterm children. Executive attention emerges towards the end of the first year of life and continues to develop at least into early adolescence [58]. Especially delayed response, on which the executive attention task in our study was solely based, continues to develop after 18 months. Indeed, previous research indicates differences in executive attention appear between children born moderately preterm and full-term as they grow older [55,59]. Thus, the task we used to asses executive attention may not have covered meaningful differences at age of assessment, that could have had an effect on later language functioning. At older ages, it is furthermore likely that executive attention is related to language functioning [60], although this effect was not found in another low-risk premature sample [61]. In short, the three subdomains of attention are not equally important in the relation between gestational age and language functioning and should be studied separately when examining attention in relation to language functioning. Future research is needed to explore the underlying mechanisms. 

While we did find that maternal sensitivity was directly related to expressive language functioning, maternal sensitivity did not mediate the association between gestational age, attention capacities, and language functioning. That is, gestational age was not related to maternal sensitivity: Mothers of children born at a lower gestational age were not more or less sensitive in interaction with their child at 18 months (corrected age) compared to mothers of children born at a higher gestational age. This finding is in agreement with two previous studies in children born very preterm [19,31]. However, children whose mothers were more sensitive in interaction showed better expressive language functioning at 24 months (corrected age). In agreement with general ideas concerning socioenvironmental influences on language development [27], this finding indicates that how mothers behave in interaction with their child and the content and quality of the communicative experiences they provide are important in the child’s language development (regardless of gestational age). 

We thus found differential relations for receptive and expressive language, that is, gestational age was only related to receptive language skills, while maternal sensitivity was only related to expressive language skills. It may be argued that expressive language is a reflection of several developmental domains (e.g., executive control, behavioral inhibition, negative affect, personality [56,62]). For example, a child who is shyer may be less likely to speak in an expressive language test (such as used in the current study) than a child who is less shy. Indeed, Salley and Dixon [56] found that 21-month-old children who had low executive control and were more fearful or frustrated showed lower expressive language skills. Receptive language may be less affected by such factors, as was shown in a study by Smith Watts et al. [62], who found that behavioral inhibition was bidirectionally related to expressive language from toddlerhood to middle childhood, but unrelated to receptive language functioning. Increased maternal sensitivity at toddler age may reflect higher responsiveness in appropriate language stimulation [29]. Maternal sensitivity then promotes expressive language functioning, even in shy children [63], which helps children in performance, but not in capability [62]. Maternal sensitivity thus provides children with motivation to express themselves more easily but not necessarily to internalize language. This may explain the absence of a relation between maternal sensitivity and receptive language. However, further research is needed to explore these mechanisms.

Moreover, we also found no support for a mediating role of maternal sensitivity in the relation between gestational age and attention capacities. The lack of a relation between maternal sensitivity and attention capacities is not in agreement with previous research that found maternal warmth and responsiveness supported alerting and executive attention functioning in full-term toddlers [34,35]. Our sample consisted of mostly highly educated, well-functioning mothers who all displayed relatively high levels of and little variation in sensitivity. Thus, maternal sensitivity in our study could perhaps generally be considered as sufficient, possibly accounting for the absence of an effect. Indeed, a study in which no relation between responsiveness and executive functioning was found showed that, on average, mothers also exhibited relatively high sensitivity [64]. Furthermore, as many previous studies indicate high educational level is related to more sensitive parenting behavior [65,66], this may further explain the absence of a relation between maternal sensitivity and attention capacities in our study. Moreover, the large proportion of highly educated mothers in our sample may also explain the absence of a relation between maternal educational level and language functioning, as generally children from low-education backgrounds are at increased risk for poorer language functioning compared to their peers from high-education backgrounds [47]. It is likely that effects of maternal sensitivity and maternal educational level are found in more heterogeneous samples.

The results of this study fit Sameroff’s Unified Theory of Development [2]. Multiple developmental systems at various levels are intertwined in their influence on early child development. In addition, it seems that biological factors such as gestational age and elementary cognitive aspects in development such as attention—a child characteristic—are more important than socioenvironmental influences such as maternal sensitivity for the development of language comprehension, i.e., receptive language. Maternal sensitivity may be more important for speech in general, i.e., expressive language. Although expressive and receptive language functioning were highly correlated, they are differentially related to biological risk, the child’s attention capacities, and maternal sensitivity. These results highlight the importance of considering receptive and expressive language as separate constructs when assessing language functioning in research and practice during the toddlerhood.

### 4.1. Limitations and Strengths

Some limitations should be noted. First, our findings should be interpreted carefully, as the scores children obtained in language functioning were largely within the normal range and the effect sizes were only small. However, small effect sizes may still be important, as they could be indicators of later, more serious problems. A further limitation of the current study lies in the generalizability of results, as a relatively large group of mothers in our sample was highly educated. Future research may attempt to explore the models of the current study in a more demographically diverse sample. 

Despite these limitations, a particular strength of our study is that we studied an integrated model of biological factors, child attention capacities, and maternal behavior in relation to language functioning at 24 months in a longitudinal, multimethod design. Previous studies often focused on a single factor or on the separate influence of multiple factors. Focusing on a combination of factors reveals the relative importance of these factors. 

### 4.2. Clinical Implications

The current study highlights that children born at a lower gestational age are at risk for poorer language comprehension early in life. Clinicians aiming to prevent problems or improve language functioning may screen moderately preterm children for language functioning, but also for alerting attention capacities. If a child already shows problems in alerting attention capacities before 2 years of age, implementing existing treatment strategies targeting attention problems or language functioning support may help prevent further problems. Additionally, as our results suggest that higher rates of maternal sensitivity promote language production, fostering maternal sensitivity in interaction between mother and child may help support development of speech. 

### 4.3. Future Research

The results of the current study fit with Sameroff’s theory [2] well. Still, other important factors affecting language functioning need to be explored. Given that alerting attention only partially accounted for differences in receptive language functioning, and gestational age, attention and maternal sensitivity explained a relatively small amount of variance, other factors also play a role in language development. Future studies could examine the contribution of other cognitive functions, like working memory [67], and other skills of children, like motor development, in relation to language development [68]. Furthermore, it could be important to examine additional environmental factors in moderately premature samples, such as maternal intrusiveness [19], paternal behavior [69], and quantity and quality of language input a child receives [70]. Moreover, it is important to investigate how our findings extend into language functioning at school age. 

## 5. Conclusions

This study assessed a combination of factors in an integrated, longitudinal, multimethod model, which allowed us to consider the relative importance of risk factors. We identified several factors that promote a healthy language development in toddlers. Gestational age, attention capacities, and maternal sensitivity all showed relative importance, although some factors were more important than others. Whereas maternal sensitivity was related to speech in toddlerhood, alerting attention was associated with gestational age and poorer language comprehension. As language comprehension and speech in toddlerhood show different associations with biological, child, contextual, and regulation factors, they should be viewed as separate constructs in research and practice.

## Figures and Tables

**Figure 1 ijerph-17-07926-f001:**
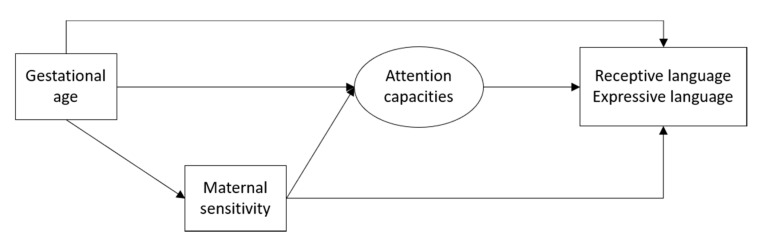
Conceptual model.

**Figure 2 ijerph-17-07926-f002:**
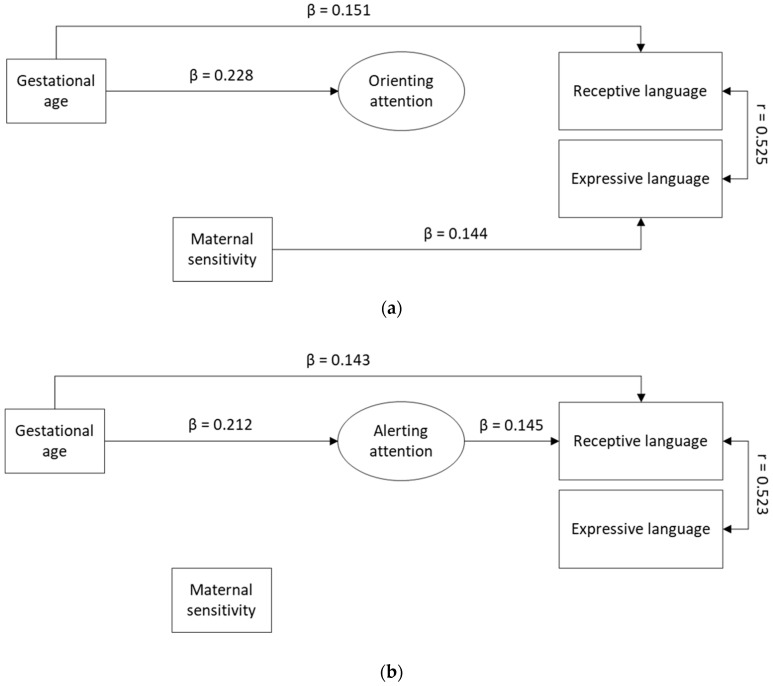
(**a**). Results of the final model for orienting attention. Non-significant paths are omitted. (**b**). Results of the final model for alerting attention. Non-significant paths are omitted. (**c**). Results of the final model for executive attention. Non-significant paths are omitted.

**Table 1 ijerph-17-07926-t001:** Demographic and neonatal characteristics of participants.

Variable		Total Sample Gestational Age 32–41 Weeks, *n* = 221	
		*M* (*SD*)	*n* (%)
Age at first assessment in months		17.93 (0.27)	
Age at second assessment in months		24.04 (0.46)	
Gestational age in weeks		36.85 (2.67)	
	32 weeks		12 (5.4)
	33 weeks		14 (6.3)
	34 weeks		20 (9.0)
	35 weeks		30 (13.6)
	36 weeks		45 (20.4)
	37 weeks		4 (1.8)
	38 weeks		11 (5.0)
	39 weeks		32 (14.5)
	40 weeks		40 (18.1)
	41 weeks		13 (5.9)
Birth weight in grams		3033.54 (692.50)	
Boys			114 (51.6)
First born			128 (57.9)
Ethnic origin child (Dutch)			212 (96.4)
Maternal nationality (Dutch)			210 (95.5)
Maternal educational level			
	Low ^a^		14 (6.4)
	Medium ^b^		54 (24.5)
	High ^c^		152 (69.1)
Marital status (married/living together)			217 (98.6)

^a^ No education, elementary school, special education, or lower general secondary education. ^b^ High school or vocational education. ^c^ College, university, or higher.

**Table 2 ijerph-17-07926-t002:** Means, SDs of, and correlations between model variables.

Variable	1.	2.	3.	4.	5.	6.	7.	8.	9.
1. Gestational Age	1								
2. Maternal Sensitivity	0.019	1							
3. Orienting Attention	0.227 **	0.040	1						
4. Alerting Attention	0.211 **	0.123 *	0.794 **	1					
5. Executive Attention	0.050	0.077	0.387 **	0.573 **	1				
6. Receptive Language	0.183 **	0.108	0.141	0.180 *	0.031	1			
7. Expressive Language	0.066	0.135 *	0.082	0.097	0.070	0.568 **	1		
8. Educational Level	0.302 **	0.080	0.262 **	0.241 **	0.055	0.090	0.110	1	
9. Birth Order	0.086	−0.002	0.067	0.065	−0.001	−0.083	−0.207 **	−0.026	1
*M*	36.85 ^a^	4.52	0.00 ^b^	0.00 ^b^	0.00 ^b^	11.50 ^c^	11.30 ^c^	–	–
*SD*	2.67	0.39	0.43	0.61	0.69	2.69	2.53	–	–
Min	32	3.14	−1.47	−1.56	−2.00	3	5	–	–
Max	41	5.00	1.31	1.09	2.06	19	16	–	–

Note. Receptive and expressive language represent language scores corrected age in case of prematurity. ^a^ Weeks. ^b^ Standardized scores. ^c^ Norm score corrected age. * *p* < 0.05. ** *p* < 0.01.

**Table 3 ijerph-17-07926-t003:** Model fit indices.

Model		BIC	*χ* ^2^	*df*	RMSEA	CFI	TLI
Orienting Attention							
	Model 1a	2464.25	10.498	6	0.058	0.961	0.857
	Model 1b	2454.85	11.887	8	0.047	0.966	0.907
Alerting Attention							
	Model 2a	2600.74	9.037	6	0.048	0.974	0.905
	Model 2b	2591.38	10.463	8	0.037	0.979	0.942
Executive Attention							
	Model 3a	2668.80	2.764	6	0.000	1.000	1.123
	Model 3b	2659.64	4.394	8	0.000	1.000	1.103

Note. a = model controlled for maternal educational level, gender and birth order. b = model controlled for gender and birth order. Model fit indices for models using language scores corrected age in case of prematurity.

**Table 4 ijerph-17-07926-t004:** Parameter estimates of the final models.

Variable		*B*	*SE* of *B*	Beta	95% CI	
					*LL*	*UL*
Orienting Attention						
	GA → sensitivity	0.003	0.010	0.019	−0.016	0.022
	GA → orienting	0.037	0.012	0.228	0.014	0.059
	GA → receptive	0.151	0.067	0.151	0.018	0.283
	GA → expressive	0.049	0.059	0.052	−0.066	0.164
	Sensitivity → orienting	0.043	0.079	0.039	−0.111	0.197
	Sensitivity → receptive	0.782	0.488	0.115	−0.096	1.660
	Sensitivity → expressive	0.921	0.457	0.070	0.026	1.816
	Orienting → receptive	0.638	0.391	0.104	−0.129	1.281
	Orienting → expressive	0.415	0.397	0.071	−0.363	1.067
	Gender → receptive	0.994	0.340	0.187	0.327	1.661
	Gender → expressive	0.820	0.336	0.163	0.160	1.479
	Birth order → receptive	−0.374	0.250	−0.097	−0.865	0.117
	Birth order → expressive	−0.760	0.237	−0.209	−1.225	−0.295
Indirect effects						
	GA → orienting → receptive	0.024	0.016	0.024	−0.009	0.056
	GA → orienting → expressive	0.015	0.016	0.016	−0.015	0.046
	GA → sensitivity → orienting	0.000	0.001	0.001	−0.002	0.002
	GA → sensitivity → orienting → receptive	0.000	0.001	0.000	−0.001	0.001
	GA → sensitivity → orienting → expressive	0.000	0.000	0.000	−0.001	0.001
Alerting Attention						
	GA → sensitivity	0.003	0.010	0.019	−0.019	0.020
	GA → alerting	0.048	0.015	0.212	0.022	0.079
	GA → receptive	0.143	0.068	0.143	0.007	0.277
	GA → expressive	0.047	0.058	0.050	−0.074	0.157
	Sensitivity → alerting	0.190	0.120	0.122	−0.038	0.431
	Sensitivity → receptive	0.689	0.452	0.101	−0.167	1.561
	Sensitivity → expressive	0.875	0.465	0.137	−0.029	1.825
	Alerting → receptive	0.638	0.289	0.145	0.014	1.195
	Alerting → expressive	0.339	0.284	0.082	−0.227	0.878
	Gender → receptive	1.016	0.339	0.191	0.347	1.663
	Gender → expressive	0.833	0.336	0.166	0.210	1.533
	Birth order → receptive	−0.373	0.249	−0.097	−0.842	0.135
	Birth order → expressive	−0.758	0.236	−0.209	−1.214	−0.300
Indirect effects						
	GA → alerting → receptive	0.031	0.017	0.031	0.005	0.075
	GA → alerting → expressive	0.016	0.015	0.017	−0.002	0.005
	GA → sensitivity → alerting	0.001	0.002	0.002	−0.003	0.007
	GA → sensitivity → alerting → receptive	0.000	0.001	0.000	−0.008	0.053
	GA → sensitivity → alerting → expressive	0.000	0.001	0.000	−0.001	0.004
Executive Attention						
	GA → sensitivity	0.003	0.010	0.019	−0.019	0.020
	GA → executive	0.013	0.016	0.049	−0.017	0.045
	GA → receptive	0.173	0.066	0.174	0.048	0.307
	GA → expressive	0.061	0.058	0.065	−0.058	0.176
	Sensitivity → executive	0.136	0.139	0.077	−0.115	0.417
	Sensitivity → receptive	0.808	0.459	0.119	−0.071	1.712
	Sensitivity → expressive	0.915	0.463	0.143	−0.271	0.700
	Executive → receptive	0.064	0.257	0.017	−0.489	0.501
	Executive → expressive	0.209	0.248	0.057	−0.271	0.700
	Gender → receptive	1.002	0.342	0.188	0.325	1.681
	Gender → expressive	0.827	0.336	0.165	0.204	1.551
	Birth order → receptive	−0.356	0.252	−0.092	−0.844	0.131
	Birth order → expressive	−0.747	0.237	−0.206	−1.198	−0.290
Indirect effects						
	GA → executive → receptive	0.001	0.005	0.001	−0.006	0.017
	GA → executive → expressive	0.003	0.006	0.003	−0.003	0.024
	GA → sensitivity → executive	0.000	0.002	0.001	−0.003	0.006
	GA → sensitivity → executive → receptive	0.000	0.000	0.000	−0.001	0.002
	GA → sensitivity → executive → expressive	0.000	0.001	0.000	0.000	0.002

Note. GA = gestational age; CI = confidence interval; LL = lower limit; UL = upper limit. 95% bias-corrected bootstrapped confidence intervals indicate significance of the coefficients in the model when CIs contain no zero. Parameter estimates reported of models using language scores corrected age in case of prematurity.

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
