# Peer review of "Toddlers’ Language Development: The Gradual Effect of Gestational Age, Attention Capacities, and Maternal Sensitivity"

_ijerph, 2020, doi:10.3390/ijerph17217926_

Round 1
Reviewer 1 Report
This manuscript explores whether infants born premature are at risk for receptive and expressive language problems at 24 months of age (corrected). Specifically, the authors investigated how gestational age, attention capacities, and maternal sensitivity related to language development. They present data from 221 infants and concluded that gestational age was positively related to receptive language and attention orienting and alerting attention. Moreover, maternal sensitivity was positively directly related to expressive language functioning. One strength of this study is use of the integrated model including biological, cognitive and quality of maternal behavior. The authors included multi-method and longitudinal data collection methods. Finally, the data highlight the importance of looking at language functioning separating for comprehension difficulties and speech difficulties. One area for improvement in the current study is a deeper discussion of alternative explanations. I think the authors described the strengths and the significant contribution of this study well and I agree that it will make a meaningful contribution to the field.
Reviewer 2 Report
This study looked at the role of prematurity, maternal responsiveness during mother-child interactions and attention at 18-months of age on children’s language scores at 24-months of age in a sample of 221 in the Netherlands. Measures used include an eye tracking measure of attention, mother-child interactions and the Bayley. Children’s gestational age ranged from moderately preterm to full term. Overall, the authors found that gestational age was related to receptive language and orienting and alerting attention, with alerting attention explaining part of the relationship between gestational age and receptive language. In addition, maternal sensitivity was related to expressive language. The authors suggest that the implications of these findings highlight the importance of screening for attentional problems early on to address potential issues for language development.
This is an interesting paper with important findings for language development, however I don’t have enough expertise with the statistical methods used to comment on the analyses.
Minor comments:
How long did the UTATE take? Were the attention tasks counterbalanced?
How long did the Bayley take?
Line 189 missing bracket.
In the results you discuss birth order but it needs to be clearer that the information you have on this is whether the child was first born or not rather than first, second, third, fourth etc.
The discussion for finding no relationship between maternal education and language scores is missing and important to consider.
Reviewer 3 Report
Overall, this is a well-written manuscript and a well-executed study. However, I believe the manuscript would benefit from some minor revision and added detail. A summary of my recommendations are outlined below.
- In the introduction, the authors should more clearly link gestational age to one of the four factors outlined in the theoretical model motivating their study.
- One of the primary strengths of the study was the separation of attentional and language skills into sub-component processes. This approach allowed for a more nuanced investigation of the factors that support early language development and resulted in a complex but interesting pattern of results. In the discussion, the authors should include more discussion around why their specific pattern of results was observed. For example:
- Why was gestational age only related to receptive language skills and maternal sensitivity only related to expressive language skills?
- Why did alerting attention show the strongest relation to language skills, as compared to the other two attentional components?
- The authors predict several indirect relations that were not supported by the data. In the discussion, the authors should provide more discussion around why these indirect effects may not have emerged in their data.
- Page 2, Line 45: There is a missing closing parentheses after the bracket
- Page 5, Line 168: The text after the parentheses can be removed as it is already included in the beginning of the sentence
- Page 6, Line 189: There is a missing closing parentheses after “…puzzle.”
- Page 9, Table 1: The name of each of the three models should be formatting similarly
- Page 13, Line 315: There is an extra opening parentheses that can be removed
- Page 15, Line 404: A comma should be added after “motor development”
- Page 15, Line 415: The word “especially” should be removed from the sentence
